# Neural Networks for Directed Connectivity Estimation in Source-Reconstructed EEG Data

Axel Faes *,†, Iris Vantieghem † and Marc M. Van Hulle

Laboratory for Neuro- and Psychophysiology, Department of Neurosciences, Medical School,
Katholieke Universiteit Leuven, 3000 Leuven, Belgium; iris.vantieghem@student.kuleuven.be (I.V.);
marc.vanhulle@kuleuven.be (M.M.V.H.)
* Correspondence: axel.faes@kuleuven.be
† These authors contributed equally to this work.

**Abstract:** Directed connectivity between brain sources identified from scalp electroencephalography (EEG) can shed light on the brain's information flows and provide a biomarker of neurological disorders. However, as volume conductance results in scalp activity being a mix of activities originating from multiple sources, the correct interpretation of their connectivity is a formidable challenge despite source localization being applied with some success. Traditional connectivity approaches rely on statistical assumptions that usually do not hold for EEG, calling for a model-free approach. We investigated several types of Artificial Neural Networks in estimating Directed Connectivity between Reconstructed EEG Sources and assessed their accuracy with respect to several ground truths. We show that a Long Short-Term Memory neural network with Non-Uniform Embedding yields the most promising results due to its relative robustness to differing dipole locations. We conclude that certain network architectures can compete with the already established methods for brain connectivity analysis.

**Keywords:** brain connectivity; artificial neural networks; source reconstruction; granger causality; time series

## 1. Introduction

A challenging problem in neuroimaging is to estimate directed connectivity between brain regions reconstructed from scalp EEG recordings but important to unveil their joint dynamics. Due to volume conduction, a given EEG electrode can pick up signals from several sources simultaneously, distorted along the way due to the presence of tissues with different electrical properties. Resolving these sources is called the "inverse problem", and it consists of estimating the source parameters given the scalp EEG recordings. The number of sources is higher than the number of electrodes, rendering an ill-posed problem. Valid brain connectivity estimation critically depends on the correct localization and time series reconstruction in this stage. Several localization methods have been proposed, often yielding differing outcomes. In a comprehensive set of simulations, [1] studied the influence of several inverse solutions, the depth of the sources, their reciprocal distance, and the Signal-to-Noise Ratio (SNR) of the recordings. They found that all these factors had a significant impact on the resulting connectivity pattern and that the number of spurious connectivity estimations depends heavily on the combinations of these factors.

In addition to the said factors, the choice of the connectivity estimator also has a significant impact. Our interest lies in directed connectivity estimation, of which partial directed coherence [2], dynamic causal modeling [3], structural equation modeling [4] and (conditional) Granger causality (GC) [5] are well-known methods. However, they rely on statistical assumptions that usually do not hold for EEG data, such as linearity [6], stationarity and prior assumptions on connectivity being expressible as a relation between

time series. However, even though some of these assumptions are violated, these methods still are best practice cases of directed connectivity estimation. In what follows, we focused on variations in traditional Granger Causality, given that it does not rely on an a priori assumed connectivity pattern. Granger Causality is a statistical hypothesis used to determine temporal causal effects between two time series. If the past of a second time series (Z) together with the past of a first time series (Y) (i.e., the "full" model) results in an improved prediction of the future value of the first time series, then the past of the first time series alone (the "reduced" model), it is said that time series Z "Granger-causes" Y.

Two main problems with this bivariate model can be discerned. Firstly, bivariate GC does not account for other time series that may be causing both Y and Z, resulting in spurious connectivity patterns. Secondly, even when bivariate GC is extended towards multiple time series by conditioning on these other variables, it is still possible that the found influence is actually caused by a linear mixture of non-interacting sources. This is because the signal measured from one electrode usually contains contributions of several sources [7]. Important to note is the proposal of Time-Reversed Granger Causality (TRGC) by [8], further validated by [7], to reduce the impact of additive correlational noise due to source mixing. The idea is that when connectivity is based on temporal delay, directed connectivity should be reversed when the temporal order is reversed. Concretely, it is checked whether the obtained GC scores for non-reversed and reversed data have opposing directions and are both significant [1]. This is clearly different from a classical way to determine significance (i.e., a likelihood ratio test). Hence, the main difference between TRGC and traditional GC is the proposed significance procedure. Still, even with TRGC, errors in connectivity estimation are here to stay. The question remains whether a totally different approach could cope with the above-mentioned problems and could perform better, or at least equally well, in comparison with the standard approaches. Artificial Neural Networks (ANNs) were considered as particularly interesting candidates given their flexible way of approximating highly non-linear relationships between variables [9] and the fact that no a priori assumptions need to be made about signal stationarity nor the connectivity pattern (for a clear overview, see [10]). Temporal convolutional networks (TCNs), as well as recurrent neural networks (RNNs), are usually well-suited architectures for time series [11–15]. While RNNs are often seen as the gold standard for sequence modeling, TCNs have also proven their suitability, for instance, in financial forecasting [14], electric power forecasting [16] and language modeling [17]. However, it remains unclear whether ANNs can signal the presence or absence of connections and their strength. Although some authors already used ANNs to derive directed brain connectivity with multilayer perceptrons and recurrent networks [15,18], these approaches did not include source-reconstructed EEG data. As stated before, unlike EEG source reconstruction, analyses based on EEG electrode levels do not allow for trustworthy inferences about interacting regions [19]. Hence, the suitability of ANNs in deriving directed connectivity between reconstructed EEG sources remains unknown.

Our motivation to assess ANNs for directed connectivity estimation between reconstructed EEG sources was two-fold. First, although many connectivity estimators exist, it is not yet known which current ANNs architectures can cope better with source-reconstructed EEG activity and under various circumstances. The authors of [1] were the first to conduct a comprehensive simulation study on the influence of dipole location, noise level, inverse solution and connectivity estimation, as well as the interactions between these factors. It was shown that different circumstances call for different analysis pipelines and that under advanced noise levels and for particular dipole configurations, even well-established methods such as TRGC can return aberrant connectivity estimates. Second, ANNs boast several appealing modeling properties that are potentially relevant to EEG modelers, such as the ability to deal with non-stationarity, non-linearity and, depending on the ANN architecture, to dispense with the prior specification of model order.

In order to assess the ability of ANNs to correctly signal the presence or absence of directed connectivity as well as connectivity strength, we compared several ANN models,

including Conv2D, a novel ANN model we propose, with TRGC. We compared their performance for different dipole locations (i.e., Far–Deep/Far–Superficial) as this can inform us whether there is a future for ANN models in brain connectivity estimation. In addition, we evaluated the ANN models relevant for directed connectivity estimation. We investigated these issues by means of a simulation study, thereby making use of a slightly adapted version of the simulation framework developed by [1] in which we manipulated the location of the dipoles and their connectivity while keeping noise level and the choice of the inverse solution constant.

## 2. Materials and Methods

### 2.1. Simulation Procedure

The simulation framework developed by [1] was used to generate simulated EEG data originating from three dipoles. This data generating process, as well as the forward and inverse problems, were implemented in MATLAB (2020). Figure 1 shows the data generation procedure. The standard length of each generated series was 1500 time steps for Ground Truth 1 and 2.

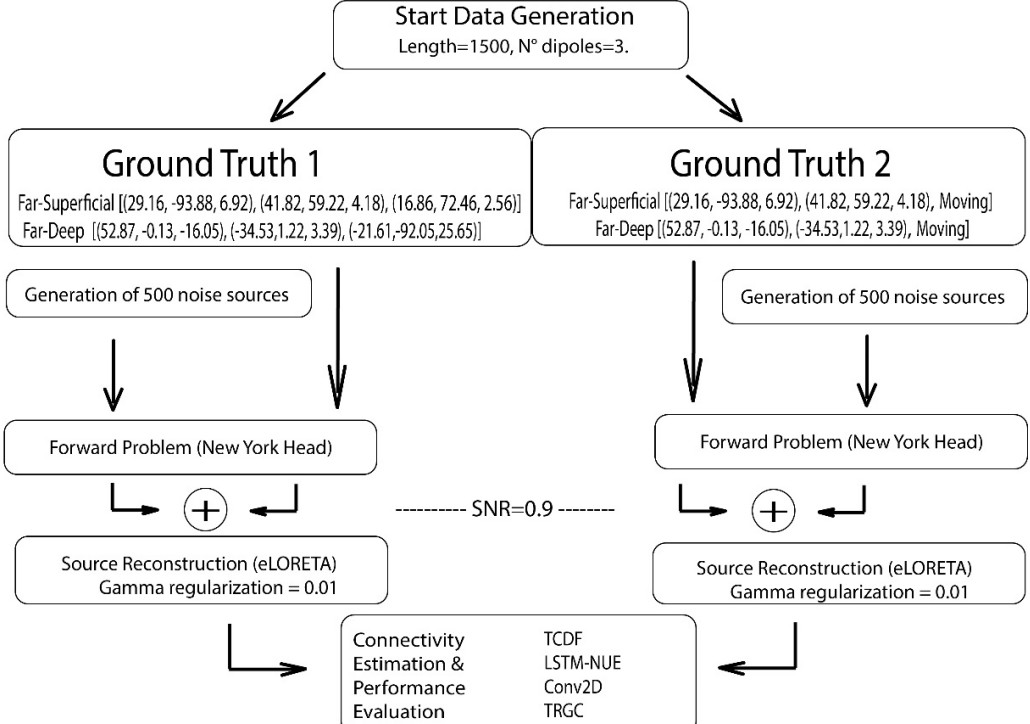

**Figure 1.** Simulation Procedure followed by Connectivity Estimation. TCDF = Depthwise Separable 1D Temporal Causal Discovery Framework, LSTM-NUE = Long Short-Term Memory with Non-Uniform Embedding, Conv2D = 2D Convolutional Network, TRGC = Time-Reversed Granger Causality. Coordinates in the Ground Truths denote MNI-coordinates.

In Ground Truth 1, three fixed dipoles were used with the directionality of the connections as well as their strength being imposed (Figure 2), a strategy used before [20–22]:

$$X_1(t) = 0.5X_1(t-1) - 0.7X_1(t-2) + c12(t)X_2 + \in_1(t)$$
$$X_2(t) = 0.7X_2(t-1) - 0.5X_2(t-2) + 0.2X_1 + c23(t)X_3(t-1) + \in_2(t) \qquad (1)$$
$$X_3(t) = 0.8X_3(t-1) + \in_3(t)$$

with $X_1$, $X_2$ and $X_3$, three electrical sources contributing to the simulated scalp-EEG signals and with:

$$c12(t) = 0.5\tfrac{t}{L} \ if \ t \ \leq \ \tfrac{L}{2} \ , \ \ c12(t) = 0.5\tfrac{L-t}{\frac{L}{2}} \ if \ t > \tfrac{L}{2}$$
$$c23(t) = 0.4 \ if \ t < 0.7L, \ \ c23(t) = 0 \ if \ t \geq 0.7L \qquad (2)$$

$L$ = length of the generated time series ($L$ = 1500), $t$ = the current time step and $\varepsilon$ = uncorrelated white noise, varying with time. We further assume an EEG cap with 108 electrodes.

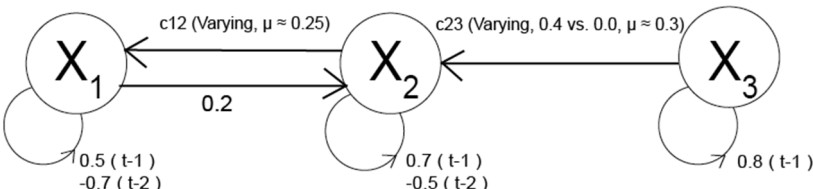

**Figure 2.** Ground Truth 1 with three fixed dipoles.

For Ground Truth 2, we considered two fixed, one moving dipole and only one true connection (Figure 3) and focused on the presence or absence of this connectivity as well as its directionality:

$$\begin{bmatrix} X_s(t) \\ X_r(t) \\ X_n(t) \end{bmatrix} = \sum_{p=1}^{P} \begin{bmatrix} a_{11}(p) & 0 & 0 \\ a_{21}(p) & a_{22}(p) & 0 \\ 0 & 0 & a_{33}(p) \end{bmatrix} \begin{bmatrix} X_s(t-p) \\ X_r(t-p) \\ X_n(t-p) \end{bmatrix} + \begin{bmatrix} \epsilon_1(t) \\ \epsilon_2(t) \\ \epsilon_3(t) \end{bmatrix} \quad (3)$$

with $X_s$, the moving dipole, as a sender, and two fixed dipoles, with $X_r$ the receiver and $X_n$ the fixed non-interactive dipole, and $a_{ij}(p)$, i, j $\epsilon$ {1, 2, 3} and $p$ $\epsilon$ {1, ... , P} the coefficients with $a_{21}$ the coupling strength between sender and receiver. All $a_{ij}$ are randomly picked from the interval [0.3, 1]. Finally, $\epsilon$ is uncorrelated, biological, white noise.

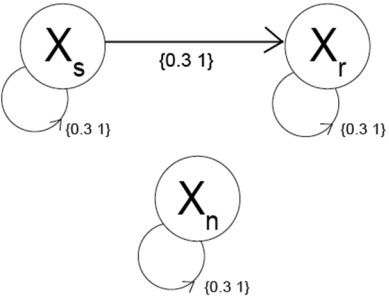

**Figure 3.** Ground Truth 2 with two fixed, one moving dipole.

The moving dipole (the sender) changes location (far, deep, close, superficial) at every iteration, with a total of 1004 iterations. The maximum time lag t is two. The reason for this ground truth is that the sender can be located at really challenging locations (too close to one of the other dipoles or very deep in the brain).

Two conditions were created for both ground truths: one condition consisted of three superficial dipoles far away from each other, while the other consisted of three dipoles located "deep" in the brain, but each dipole was still positioned far away from the other dipoles. The corresponding MNI-coordinates of the two fixed dipoles that Ground Truth 1 and 2 have in common are depicted in Figure 4. The full set of coordinates of Ground Truth 1 (including the coordinates of the third fixed dipole) is denoted in Figure 1.

In Ground Truth 2, the first two coordinates are the same as in Ground Truth 1, for each dipole condition, while the third dipole moves throughout the brain as described above. The Far–Superficial versus Far–Deep configurations indicate (relative) distances: "deep" denotes a distance from the origin (located at the anterior commissure) <6 cm and "superficial" >6.5 cm. The distance between dipoles is evaluated as "far" if the relative distance to the other dipoles exceeds 8 cm.

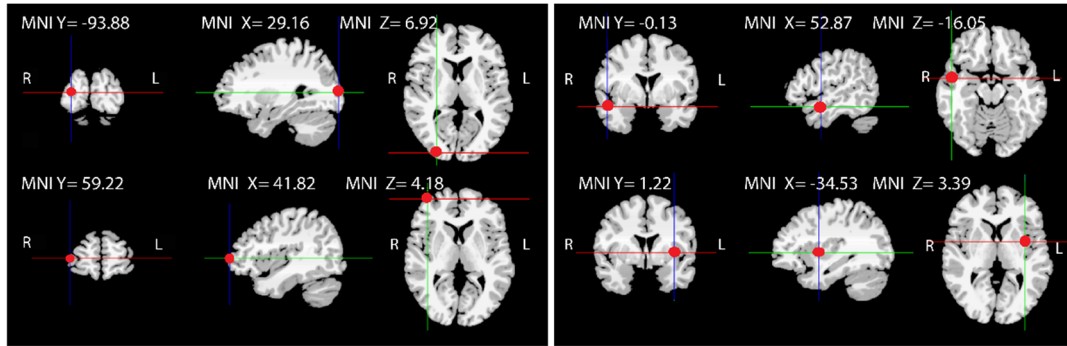

**Figure 4.** Locations (MNI-coordinates) of the dipoles the Ground Truths have in common. Left: Far–Superficial fixed dipoles. Right: Far–Deep fixed dipoles.

As an additional check for robustness of source localization, noise sources were added as a background activity. These were modeled using pink noise, also called $1/f$ noise, and created by scaling the amplitude spectrum of random white Gaussian noise with the factor$1/f$ using the Fourier transform and its inverse.

After generating these noise sources, the forward problem is construed:

$$Y = LX + e, \tag{4}$$

where $Y$ denotes the scalp-recorded potentials, $X$ represents the electrical sources in the brain (the dipoles), "$e$" is measurement noise (electrode noise) and $L$ is the head volume conductor model (also called the leadfield matrix). The leadfield matrix determines how the activity flows from dipoles to electrodes. In this work, the New York Head model [23] was used.

The pink noise and the source activity are then projected onto the scalp, after which they are summed:

$$Y^{brain}(t) = \gamma \times \frac{Y^{active}(t)}{||Y^{active}(t)||_{FRO}} + (1 - \gamma) \times \frac{Y^{noise}}{||Y^{noise}(t)||_{FRO}} \tag{5}$$

$Y^{active}$ and $Y^{noise}$ refer to the scalp-projected source signals and pink noise activity, respectively; both are scaled by dividing them by their Frobenius norm ($||Y^{active}(t)||_{FRO}$, $||Y^{active}(t)||_{FRO}$). The Signal-to-Noise Ratio (SNR) is computed for all dipoles simultaneously and set to 0.9 ($\gamma = 0.9$).

Next, white noise (spatially and temporally uncorrelated activity) is added to $Y^{brain}$ to simulate electrode noise, resulting in Equation (6) where $Y^{measurement}$ represents the simulated EEG signal. Again $\gamma = 0.9$ is imposed as Signal-to-Noise Ratio:

$$Y^{measurement}(t) = 0.9 \times \frac{Y^{brain}(t)}{||Y^{brain}(t)||_{FRO}} + 0.1 \times \frac{Y^{meas\_noise}}{||Y^{meas\_noise}(t)||_{FRO}} \tag{6}$$

Afterwards, the simulated scalp-EEG data are source-reconstructed using exact low-resolution brain electromagnetic tomography (eLORETA) [24]. There have also been improvements to eLORETA, such as Sparse eLORETA, which uses a masking approach to improve the source localization density [25]. The eLORETA method is a discrete, three-dimensional (3D), linear, weighted minimum norm inverse solution [24]. In the absence of noise, an exact zero-error localization accuracy can be obtained with eLORETA, but this does not hold for noisy data, as was shown in a study comparing both scenarios [26]. The MATLAB implementation of the eLORETA algorithm (mkfilt_eloreta2.m) from which spatial filters are obtained was developed by G. Nolte and is available in the MEG/EEG Toolbox of Hamburg (METH; https://www.uke.de/english/departments-institutes/institutes/neurophysiology-and-pathophysiology/research/research-groups/index.html, accessed

on 22 December 2021). As the input, it takes the leadfield tensor (i.e., the head model file N*M*P containing N channels, M voxels, and P dipole directions) as well as a regularization parameter gamma (set to 0.01); as the output, an N*M*P tensor A of spatial filters is returned.

### 2.2. Connectivity Models

The ANNs evaluated in this study were selected based on their suitability for time series analysis (Table 1). While TCDF outputs attention scores, for which higher scores are used to represent stronger connectivities, LSTM-NUE makes use of Granger Causality scores equaling NNGC = errreduced−errfull, which are then binarized [18]. Conv2D uses the $R^2$-score between the real and predicted values of the current target (i.e., the time steps to be predicted). While using TRGC as implemented by [1], only binary GC scores are outputted. The configuration (i.e., the used parameters) of each ANN was determined using a data-driven approach, such that for each ANN, the parameters returning the best results were chosen. This parameters pre-testing was performed with different simulated data sets (i.e., differing from the data sets that were used to report the final results).

**Table 1.** Set-up of the compared ANNs and TRGC.

| Model | Architecture | Self-Causation | Connectivity Measure |
|---|---|---|---|
| **TCDF** | Depthwise Separable 1D Convolutional Network | yes | Attention score |
| **LSTM-NUE** | Long Short-Term Memory Network | no | GC (0 or 1) |
| **Conv2D** | 2D Convolutional Network | yes | $R^2$-score |
| **TRGC** | Time-Reversed Granger Causality | no | GC (0 or 1) |

#### 2.2.1. Temporal Causal Discovery Framework

The Temporal Causal Discovery Framework (TCDF) developed by [14] is based on the concept of a one-dimensional Temporal Convolutional Network (TCN) and is available on Github [27]. Input to the framework consists of an NxL data set consisting of N time series of equal Length L. Within the framework, one depthwise-separable TCN is used to obtain a prediction for a single source (target). The input of the network consists of the history of all time series, including the target time series. The output is the history of the target time series. An attention mechanism is added: each $TCN_j$ has its own trainable attention vector $V_j = [v_{X1j}, v_{X2j}, \ldots, v_{ij}, \ldots v_{Nj}]$, that learns which of the input time series is correlated with the target by multiplying attention score $v_{ij}$ with input time series $X_i$ in $TCN_j$. When the training of the network starts, all attention scores are initialized as 1 and are, as such, adapted during training. The direction of connectivity and significance is determined using a shuffling procedure. For significance determination, one of the time series is shuffled while keeping the other one(s) intact when predicting the target. The runs with shuffled time series did not involve any model retraining. Instead, in the prediction step, the losses obtained when using the "shuffled" time series as predictors were compared with the losses obtained when using the non-shuffled time series. Only if the loss of a network increases significantly when a time series is shuffled that time series is considered a cause of the target time series. A time series X1 is only considered to be a significant contributor to another time series X2 if, in the first stage, its attention score is larger than one. Only if, after shuffling the potentially contributing time series X1, the difference between losses obtained by predicting future time steps with the unshuffled time series and losses obtained by predicting using shuffled time series is large enough, using an a priori determined threshold significance value, time series X1 is considered a significant contributor to time series X2. TCDF was run with PyTorch (version 1.4.0, www.pytorch.org, accessed on 17 December 2021).

Configuration. For TCDF, the chosen parameters were the number of hidden layers = 1, kernel size = dilation coefficient = 4 (a time-dimensional kernel), learning rate = 0.01,

optimizer = Adam, number of epochs = 1000, significance threshold= 0.9998, seed = 1000. Kernel weights are initialized following a distribution with $\mu = 0$, variance = 0.1.

### 2.2.2. LSTM-NUE—Long Short-Term Memory with Non-Uniform Embedding

Another connectivity measure is based on the RNN, in which directed cycling connections are present, i.e., there are feedback connections from output to input, and these connections create possibilities for memorization. A subtype of RNNs is the Long Short-Term Memory network (LSTM). This type of network provides a resolution for vanishing and exploding gradient problems in recurrent networks. It performs this by introducing gates and memory cells which also makes it very flexible towards gap length. The implementation in this study is an LSTM with Non-Uniform Embedding (NUE, a feature selection procedure) by [15], which is also publicly available [28]. NUE is an iterative selection procedure adopted from [18] to detect the most informative time steps of the predicting time series (phase one). In phase one, a vector $V$ containing the most informative past time steps to explain the present state of a target time series X1 is obtained by iteratively adding time steps (of the time series' own past, but also of the past of the other time series) to the training set and obtaining a new model error as a time step is added. For instance, let $V = [V^{X1}{}_n, V^{X2}{}_n, V^{X3}{}_n]$ represent the vector with the most relevant past time steps to explain the present of the target time series. This selection of time steps goes on until the prediction error becomes larger than or equal to a threshold or until the maximum amount of time steps is reached. If for a certain time series X2, no time steps have been added in $V$, the time series is not further considered as a potential contributor to target time series X1, and it is not considered in the next phase (phase two). Phase one results in an estimation of the error variance of the full model (i.e., the model containing all relevant past time steps from different time series). In phase two, the model is fit only with this smaller set of time steps. The error of the reduced model is finally obtained by not using the values of the time series (e.g., X3) that is a potential contributor to the target time series X1. If the error (Loss$^{\text{Reduced}}$) of this reduced model is larger than the error of the full model (Loss$^{\text{Full}}$), time series X3 is considered a significant contributor to time series X1 ("X3 Granger-predicts X1").

In LSTM-NUE, no shuffling is used to determine connectivity. Instead, the significance procedure consists of two phases. Determining significance is based on (1) the selection of relevant time samples from all time series rendering a full model, after which the time series whose time samples were not selected are already as potential causes of the target time series. (2) The remaining candidates are then, as a test, subsequently excluded from the model to obtain the reduced model (i.e., the model with only the target time series as its own predictor). Hence, this exclusion phase is, to some extent, comparable with the shuffling procedure used in TCDF, given that this procedure is in this way testing the relevance of a certain time series in the prediction of another (by excluding it OR by shuffling the values).

Configuration: for LSTM-NUE, the parameters are the number of hidden layers = 1, the number of units in each layer = 30, batch size = 30, num_shift = 1, sequence_length = 20, number of epochs = 100, theta = 0.09, learning rate = 0.001, weight decay = $1 \times 10^{-7}$, min_error = $1 \times 10^{-7}$ (=a priori determined error to determine whether a certain time step should be included in the final model), and train/validation split = 0.85/0.15. Default kernel initializer = "glorot_uniform", which draws samples from a uniform distribution, is used to initialize the weights of the LSTM-layer.

### 2.2.3. Conv2D—Two-Dimensional Convolutional Network

Finally, we propose a two-dimensional Convolutional Network (Conv2D) as a way to test whether a 2D kernel variation in TCDF has merit. The input consists of an NxL data set, which is transformed into a four-dimensional tensor (time samples of training set, window size, amount of predicting time series, 1). The source code is accessible via Github (kul-EEG-sourceconnectivity, https://github.com/irisv440/kul-EEG-sourceconnectivity, accessed on 21 September 2021).

Some important differences with TCDF are the fact that a two-dimensional kernel is used and that a cross-validation procedure, adapted for time series, is embedded in the framework. While in TCDF, a one-dimensional kernel (with height = 1) slides over the data along the time dimension (=width of the kernel, i.e., the amount of time steps considered together), in Conv2D, a two-dimensional kernel is used in which the second dimension represents the amount of time series that will be convolved together. The second dimension has an upper bound, which is the total amount of time series within the input data. We hypothesized that by adding a second dimension (feature dimension) to TCN, we could capture the most important aspects of the other time series, leading to more correct connectivity estimates. However, it was suggested (e.g., [29]) that convolving data from several time series can also cause less accurate results (in our case, this means lower Sensitivity and lower Precision) because too many time series are convolved together, possibly erasing the impact of changes in individual time series. Similar to TCDF, the input to the network consists of all time series, including the target time series. The output is a single target time series.

A second difference is cross-validation (CV) for time series. Cross-validation is a powerful method for detecting overfitting, but its implementation in time series models is not trivial, given that no leakage from future to past may exist. This issue was solved by using 6-fold cross-validation on a rolling basis based upon "TimeSeriesSplit" from the model selection module of the sklearn-library version 0.24.1 (Scikit-learn, original version released by [30]). With TimeSeriesSplit, we obtained the following train-test regime for the folds where "—"represents the unused part of the data in the corresponding fold (Figure 5).

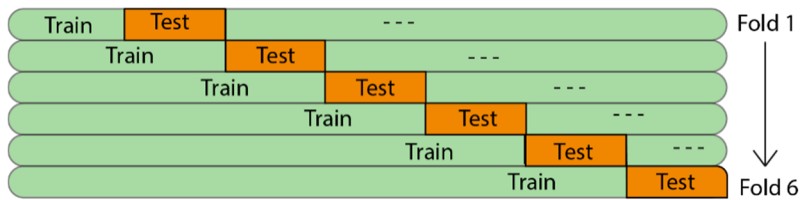

**Figure 5.** CV with length of first train-fold = length of first test-fold (= 1500/7).

In addition, given that connectivity may vary over longer time spans (as is also the case in Ground Truth 1), working with only one division in the train/validation/test-set (respecting past versus future) can cause false positives or false negatives since one may be training on a portion of the time series where connectivity is very strong between, for instance, X3 and X2 while validating and/or testing on a part where the same connectivity is weak (or the other way around).

As a metric for connectivity strength, the $R^2$-score between the real values and the predicted values of the current target is used. The better a time series pair is successful in predicting a target, the larger the similarity between the true values and the predicted values will be, hence the stronger the connectivity between the time series and target. When, for instance, two different pairs of time series X1 and X2, versus X1 and X3 are used as predictors for X1, $R^2$ again represents the similarity between predicted and true values of the target X1. When the prediction of X1 becomes better when predicted by time series X1 and X2 together, instead of with X1 and X3, one could conclude that connectivity is stronger between X1 and X2 than between X1 and X3. The $R^2$ scores themselves are obtained from the cross-validation folds, after which the average $R^2$ score is taken over the folds and over the number of used runs for one data set. The corresponding output is a scoring matrix representing all combinations of time series used as predictors and possible target time series. If the $R^2$ score is >0 and the predicting time series are considered significant (see "Connectivity Analysis using ANNs"), the obtained $R^2$ score can be interpreted. However, when including more than two predictors, this relationship is not so easily established anymore, given that the $R^2$-score still represents the connection between the target and all predicting time series together. Similar to TCDF, the direction of connectivity and

significance is determined using a shuffling procedure. Significance weights are obtained by comparing training and test loss differences, after which a data-driven cutoff (here 0.70) is used to differentiate between contributing and non-contributing time series. More concretely, training difference = (first training loss)–(final training loss), where the latter is expected to be much lower than the first term, and Test difference = (first training loss)–(loss of test-indices using shuffled train data) where the latter is expected to be high because of the shuffled data; hence, one expects the test difference to be very small. Next, if the average test difference was larger than the average_training_difference * significance (=0.9998), the potential connection is considered not significant in the first place. Significance weights are obtained by (test difference/training difference). If the weight is larger than the cutoff (=0.70), the connection is considered not significant. The used significance level, as well as the cutoff for significance weights, were experimentally determined, and the final choice was based upon a data-driven approach (by experimenting with significance levels in the range of {0.70, 1} and with cutoff-scores in the range of [0.40, 0.70]). For the current kind of simulated data, these values worked well.

Configuration: for Conv2D, the parameters were as follows: number of hidden layers = 1, number of filters = 24, kernel size = {4*2, 4*3} (width*height), dilation coefficient = 1, number of epochs = 12, window size = 5, learning rate = 0.005, optimizer = "Adam", significance = 0.9998, cut-off scores for significance weights = 0.70 and number of train/test splits for CV = 6. Default kernel initializer = "glorot_uniform", was used to initialize the weights of Keras' Conv2D-layer.

### 2.2.4. TRGC—Time-Reversed Granger Causality

As our baseline method, Time-Reversed Granger Causality (TRGC), as implemented (by means of the Matlab function "tr_gc_test", embedded in "simulation_source_connectivity"), and evaluated by [1], was used. As stated before, the difference with "traditional" GC is the type of significance procedure. Instead of the classical way to determine significance (a likelihood ratio test), which cannot distinguish between actual versus spurious correlations due to source mixing, it determines whether the "standard" GC scores for non-reversed and reversed data have opposing directions and are both significant. In other words, direction-flipping must occur when data are time-reversed. This is referred to as conjunction-based TRGC [7]. A drawback of GC (and hence, TRGC) is that one needs to define the model order, which is feasible when the ground truth is known, such as in simulations, but in "real" EEG data, this quickly becomes a tricky problem. An advantage, on the other hand, is the fact that with TRGC, one model for all sources is constructed, after which one threshold is applied to all obtained GC scores.

Configuration. Function tr_gc_test takes as input an NxL matrix H', the model order, the number of time steps in the time series, alpha, the type of significance test ("conservative", requiring significant GC scores with original as well as reversed data; versus a significance test based on difference scores between GC scores in normal and reversed order) and finally, the type of VAR model estimation regression mode to calculate pairwise-conditional time-domain Granger Causality scores. In this work, the model order of TRGC was set to two, we opted for "conservative" significance testing, and ordinary least squares (OLS) was used as Vector-Autoregression (VAR) estimator. We used an alpha level of 0.05, FDR corrected [31]. The corresponding p-value was taken as a threshold to binarize connectivity scores.

### 2.3. Performance Evaluation

The main question is whether the connections in the ground truths could be detected by the evaluated networks and by TRGC ("True Positives", TP) without detecting too many false connections ("False Positives", FP), thus connections that are not present in the ground truths. Measures based upon these are Precision, Sensitivity/Recall, and F1-score (Figure 6), which we used for comparing TCDF, LSTM-NUE, Conv2D and TRGC.

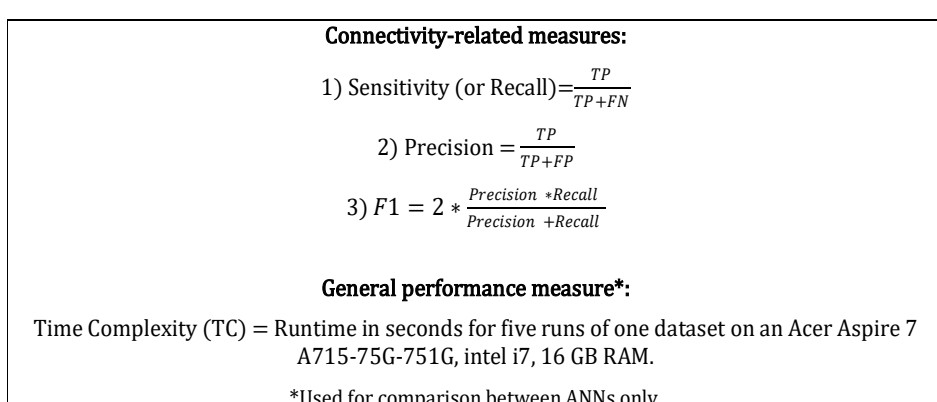

**Connectivity-related measures:**

1) Sensitivity (or Recall) $= \frac{TP}{TP+FN}$

2) Precision $= \frac{TP}{TP+FP}$

3) $F1 = 2 * \frac{Precision *Recall}{Precision +Recall}$

**General performance measure\*:**

Time Complexity (TC) = Runtime in seconds for five runs of one dataset on an Acer Aspire 7 A715-75G-751G, intel i7, 16 GB RAM.

\*Used for comparison between ANNs only.

**Figure 6.** Main evaluation measures.

The results on connectivity strength are not directly compared between models as they differ substantially. These strength estimates, based on the mean over five runs on the same data set, are calculated and ranked. It must be emphasized that these strength estimates are relative per model and target training as, for each target time series, the network is trained differently. The latter implies that connection strengths obtained in the prediction of a particular Target time series X1 cannot be readily compared with connection strengths obtained in the prediction of another target time series X2. If F1 < 50%, only rankings are presented. Self-connectivity is not taken into account to avoid an overly positive perception of the results.

## 3. Results

### 3.1. Ground Truth 1

#### 3.1.1. Connectivity Detection

In Figures 7 and 8, respectively, Sensitivity and Precision are shown per method. Some remarks, specifically with regard to Conv2D, need to be made before interpreting the results.

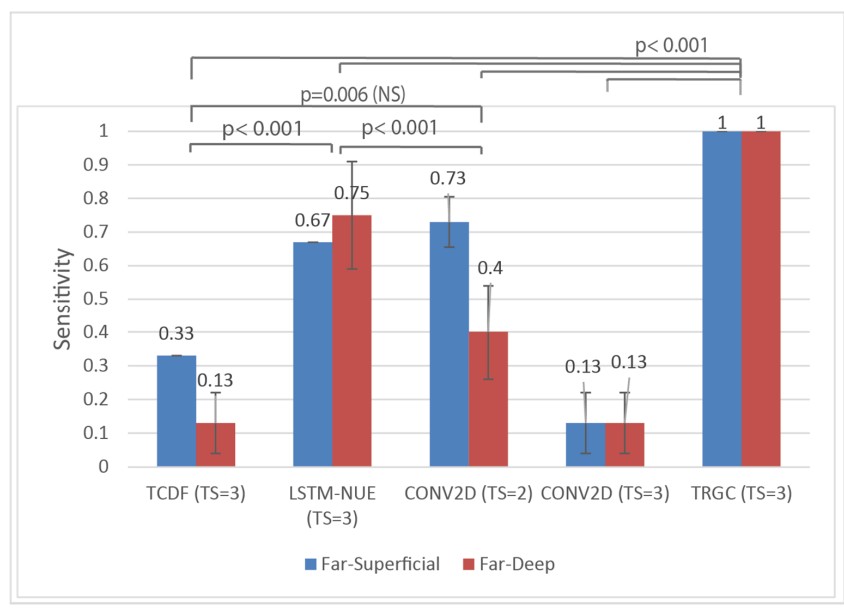

**Figure 7.** Mean Sensitivity for all methods (L = 1500). Sensitivity Ranking Far–Superficial: TRGC > Conv2D (TS = 2) > LSTM-NUE > TCDF > Conv2D (TS = 3). Sensitivity Ranking Far–Deep: TRGC > LSTM-NUE > Conv2D (TS = 2) > Conv2D (TS = 3) = TCDF. Abbreviation TS = amount of time series included in the predictions.

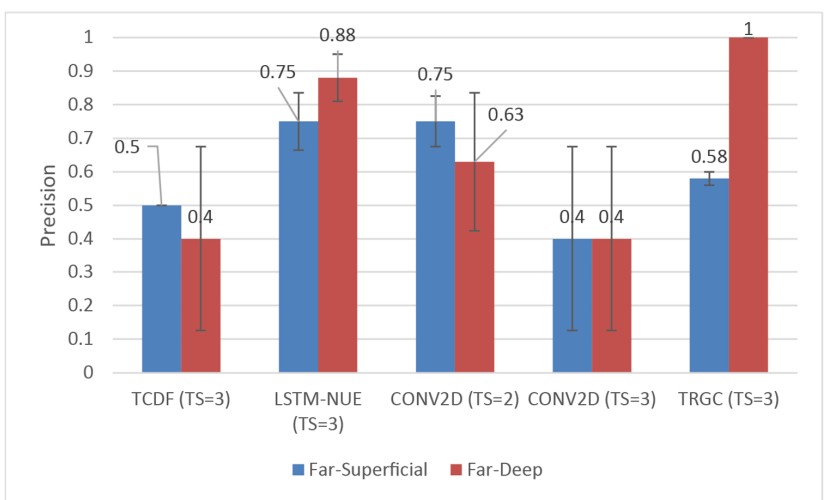

**Figure 8.** Mean Precision for all methods (L = 1500). Sensitivity Ranking Far–Superficial: TRGC > Conv2D (TS = 2) > LSTM-NUE > TCDF > Conv2D (TS = 3). Sensitivity Ranking Far–Deep: TRGC > LSTM-NUE > Conv2D (TS = 2) > Conv2D (TS = 3) = TCDF. Abbreviation TS = amount of time series included in the predictions.

Given that the Conv2D-model based on three predicting time series resulted in a very low Sensitivity (0.13 ± 0.18) and low Precision (0.40 ± 0.55), see Figures 7 and 8, it was not considered relevant to explore the model with three predictors further in terms of connectivity strength (for connectivity strength per ANN, see Sections 3.1.2–3.1.4). This decision was supported by the results of a Scheirer–Ray–Hare Test with model and dipole condition as factors and with follow-up Mann–Whitney U tests (Bonferroni-corrected). Superior results were obtained with Conv2D models containing two predicting time series versus three predicting time series. These results can be consulted in Appendix A (Table A1).

Hence, strength rankings are explored only with the Conv2D model with two predictors (Section 3.1.4). With regard to the model based on two predicting time series, the results obtained by looking at each predictor pair (consisting of two time series) separately revealed large differences between pairs in terms of Sensitivity and Precision. We chose to take all detected connections into account while calculating our scores instead of averaging over all predictor pairs, as it could lead to biased results. This is because if it is found that X2 predicts X1 when it is predicted together with X1 but not detected when it is predicted with X2 and X3, and the discovered connection between X2 and X1 is still included in the performance scores, this increases Sensitivity but decreases Precision. The decrease in Precision then occurs because if a false positive is found by one of the two predictor pairs, it is still counted. Option 1 was chosen to put the focus more upon detection ability and exploration. Thus, it must be kept in mind that a positive detection bias exists in all our overall two-to-one performance scores of Conv2D.

While focusing on differences in Sensitivity, the following results were obtained for the used ANNs and TRGC. A Scheirer–Ray–Hare Test with model and dipole condition as factors revealed no statistically significant interaction (using alpha = 0.05) between the effects of the type of connectivity method and dipole condition ($p = 0.63$), nor the main effect of dipole condition itself ($p = 0.63$). However, a simple main effects analysis showed that the type of connectivity method does have a statistically significant effect on Sensitivity ($H$ (4,40) = 38,159, $p < 0.001$). Follow-up two-sided Mann–Whitney U tests (Bonferroni-corrected: alpha = 0.05, alpha adjusted = 0.005), carried out across dipole conditions, show significant and marginally significant differences between the following methods. Median scores (denoted as *Mdn*) are reported. In contrast to TCDF (*Mdn* = 0.33), smaller contributions of one time series to another could be detected with LSTM-NUE (*Mdn* = 0.67), $p < 0.001$. The difference between TCDF and Conv2D (*Mdn* = 0.67) with two time series as predictors was only marginally significant after correcting for multiple

comparisons, *p* = 0.006. TRGC (*Mdn* = 1), however, outperformed all ANN models in terms of Sensitivity (*p*-values denoting differences with all other methods < 0.001). Finally, while no significant difference was found between LSTM-NUE (*Mdn* = 0.67) and Conv2D (*Mdn* = 0.67) with two time series as predictors (*p* = 0.239), LSTM-NUE performed significantly better than Conv2D with three time series as predictors (*Mdn* = 0), *p* < 0.001). Rankings are described below in Figure 7 to provide qualitative comparisons. Note that in Figure 7, mean scores *M* for each dipole condition is still reported, given that the current results were obtained with small sample sizes. Hence, differences between dipole conditions may still appear once statistical power is increased (i.e., by using more data sets) and given that differences between dipole conditions were, to some extent, expected.

With regard to Precision, the results of a Scheirer–Ray–Hare Test with method and dipole condition as factors were not significant, albeit a marginally significant result for method ($H$ (4,40) = 8.79, *p* = 0.07) was obtained. Hence, no follow-up tests were carried out.

Thus, we rely upon rankings only for our qualitative description (in terms of mean scores *M*, taking dipole condition into account) of the data. In the Far–Superficial dipole condition, Conv2D with TS = 2 and LSTM-NUE obtain both a Precision of *M* = 0.75 (±0.15, 0.17, respectively), followed by TRGC (*M* = 0.58 ± 0.05) and TCDF (*M* = 0.50 ± 0). Precision is lowest in Conv2D with TS = 3 (*M* = 0.40 ± 0.55). However, in the Far–Deep dipole condition, TRGC obtains perfect Precision (*M* = 1 ± 0), followed by LSTM-NUE (*M* = 0.88 ± 0.14) and Conv2D with TS = 2 (*M* = 0.63 ± 0.41). The qualitatively lower Precision score of TRGC in the Far–Superficial condition turned out to be mainly due to two consistently observed false-positive connections that were not detected in the Far–Deep dipole condition.

When summarizing the results in terms of F1-scores, the following ranking was obtained for the Far–Superficial condition: TRGC (*M* = 0.73 ± 0.04) = Conv2D with TS = 2 (*M* = 0.73 ± 0.09) > LSTM-NUE (*M* = 0.70 ± 0.07) > TCDF (*M* = 0.40 ± 0.0) > Conv2D with TS = 3 (*M* = 0.20 ± 0.27).

For the Far–Deep condition, the F1-score ranking was as follows: 1 ± 0 (TRGC) > 0.75 ± 0.17 (LSTM-NUE) > 0.47 ± 0.31 (Conv2D with TS = 2) > 0.20 ± 0.27 (Conv2D with TS = 3) = 0.20 ± 0.27 (TCDF).

### 3.1.2. TCDF

With regard to TCDF (Figure 9), only two mean attention scores were significant, and solely in the Far–Superficial dipole condition and for two different targets (X2 and X3), such that a target-wise comparison cannot be made. More concretely, a connection X3 → X2 was found (in accordance with the ground truth), as well as a connection X2 → X3 (unlike the ground truth).

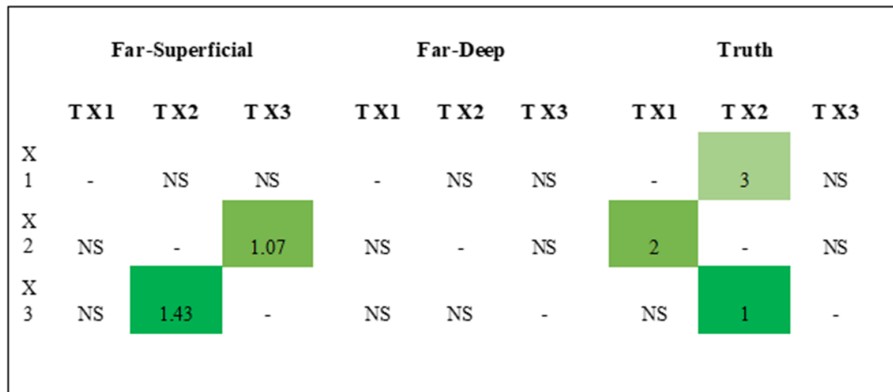

**Figure 9.** TCDF, Mean attention score rankings. Colors: "1", green, denoting the highest attention score in one TCN—one column). The columns represent the targets (T X1, T X2, T X3), the rows the predictors (X1, X2, X3). Self-connectivity is excluded.

### 3.1.3. LSTM-NUE

Next, when focusing on LSTM-NUE, as can be seen from the colors from Figure 10, for target time series X3, the GC scores (in both dipole conditions) were higher than expected according to the ground truth. Unexpected GC scores are surrounded by black rectangles in the top panel. Column-wise strength rankings (rankings for one particular target) are correct for two out of three targets (X1, X2) in both conditions, as can be seen by comparing with column-wise Ground Truth 1 (Figure 10, bottom right panel). The overall ranking in the Far–Deep condition was more in accordance with the overall ranking in Ground Truth 1 (Figure 10, bottom left panel) than the ranking found in the Far–Superficial condition because connectivity strength was observed to be the weakest for the corresponding false positives (as shown in yellow in the top panel).

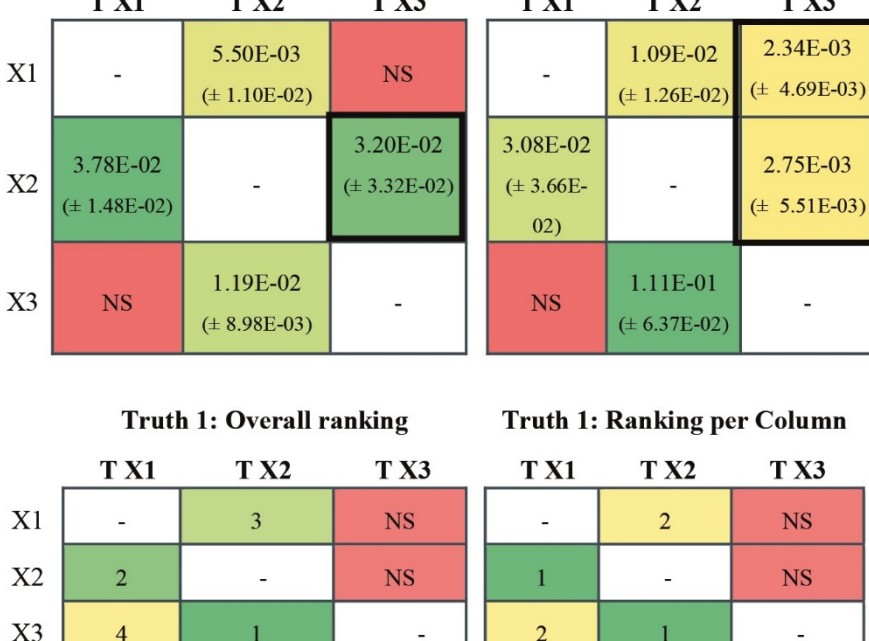

**Figure 10.** LSTM-NUE, Neural-Network Granger Scores rankings (Top) versus Truth 1 (Bottom), excluding self-connectivity. Color coding: dark green > light green > yellow > orange > red. The columns represent the targets, the rows the time series used for prediction.

### 3.1.4. Conv2D

In Figure 11, $R^2$-strength rankings for Conv2D with two time series are shown. Given that for Conv2D, adding a third time series did not work out well, only rankings per predictor pair could be obtained. When $R^2$-strength is shown in the upper two panels of Figure 11, it means that the current time series pair is a significant contributor. Significance weights, which denote significant contributions of one time series to a target time series (instead of $R^2$ scores denoting a connection between a certain pair of predicting time series and one target time series), are reported between brackets. They were obtained as described in Section 2.2.3 and were considered significant if the cutoff of 0.70 was not exceeded. The lower two panels show a ranking (with 1 being the most active connection and 4 the least active connection).

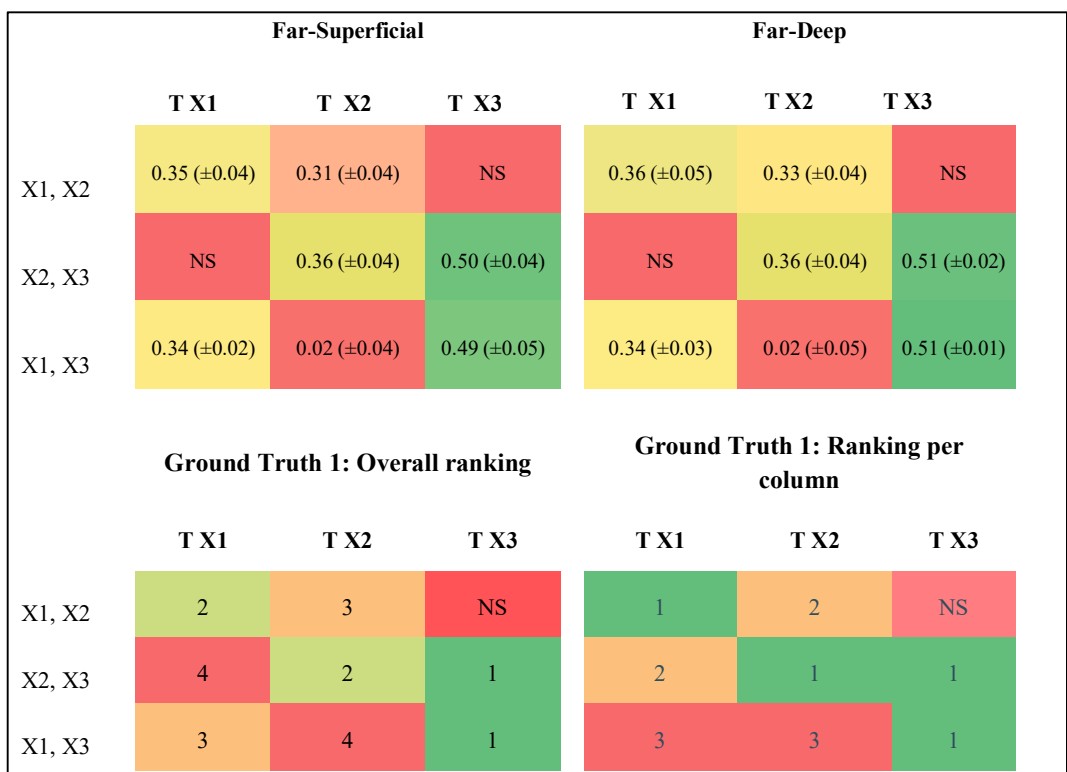

**Figure 11.** Conv2D, $R^2$-strength score for time series pairs (Top) versus ranking of connections in the Ground Truth 1 (Bottom, 1 being the strongest), including self-connectivity. Color coding: dark green > light green > yellow > orange > red. The columns represent the targets, the rows the time series pairs used for predicting the target time series.

It can be seen from Figure 11 that, in the prediction of Target X1, out of three direct connections, only one is not significant (i.e., X2, X3 → X1, Top-row), but this is only the case when Target X1 is not included as a predictor. Regarding an overall ranking (Ground Truth 1, Bottom-Left), it can be seen that X3 has the strongest self-connectivity, while for X1 and X2, self-connectivity is almost the same. This is observed in our results as well ($R^2$ = 0.36, 0.35 in both dipole conditions). Moreover, the obtained $R^2$ strength scores are not, or barely, dependent on the dipole condition. Next, while inspecting these results column-wise (hence, target-wise), a stronger connection between predictors X1, X2 and Target X1 than between predictors X1, X3 and target X1 were expected. However, these connections are quite similar ($R^2$ = 0.35 versus $R^2$ = 0.34 in the Far–Superficial condition, $R^2$ = 0.36 versus $R^2$ = 0.34 in the Far–Deep condition). While predicting target X2 using X1, X3, a significant, correct contribution from X1 to X2 is found (significance weight = 0.232, 0.120, Far–Superficial and Far–Deep condition, respectively), as well as a correct contribution from X3 to X2 (significance weight = 0.001, 0.048, Far–Superficial and Far–Deep condition, respectively). However, connectivity strength $R^2$ is very low ($R^2$ = 0.02 in both dipole conditions) in comparison with the situation in which target X2 is included in the predictor pair and in which case X3 is also considered a significant contributor (predictor pair = X2, X3, $R^2$ = 0.36,0.36, significance weights = 0.341,0.210 for X3, Far–Superficial and Far–Deep condition, respectively). The ranking for Target X2 is correct, as was the ranking for X1. Finally, we expected similar rankings for X2; X3 predicting X3 as for X1; X3 predicting X3 since neither X1 nor X2 contribute to X3. This is indeed the case for both conditions. As expected, significance weights for individual contributions of X1 and X2 to X3 were not significant (significance weights >0.70 in both dipole conditions).

### 3.1.5. Time Complexity

Finally, we assessed runtimes in seconds for one data set (including averaging over five runs) w.r.t. the training of the ANNs. Runtimes with three time series as predictors (TS = 3), Length L = 1500, dipole condition = Far–Superficial are shown in Figure 12, as well as the runtime of Conv2D with two time series as predictors. The runtime of Conv2D with only two time series as predictors was 1048 s. All runs were performed with an Acer Aspire 7 A715-75G-751G, intel i7, 16 GB RAM.

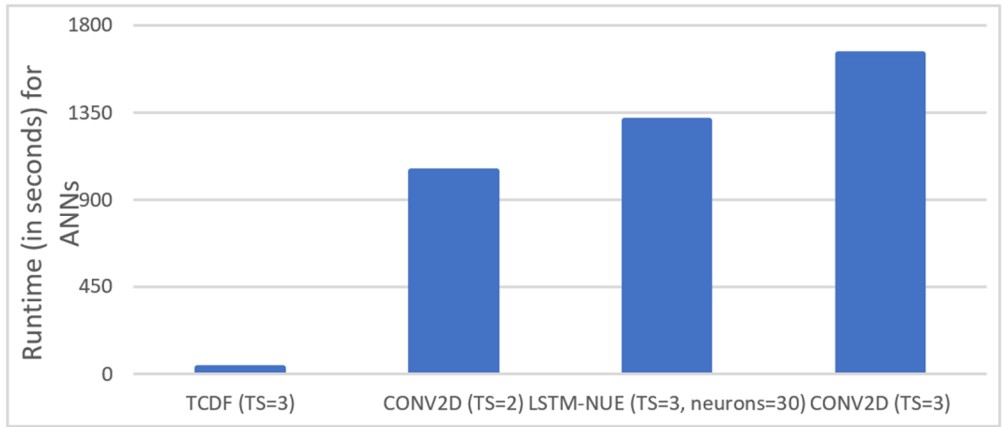

**Figure 12.** Runtime for the training of all ANNs, with 3 time series (TS = 3), L = 1500. An extra comparison showing the runtime of Conv2D with two time series as predictors (TS = 2) is shown (but all datasets contain 3 time series). "Neurons" = the number of hidden layer neurons.

### 3.2. Ground Truth 2

With only one true connection and excluding self-connectivity, it was found that none of the methods, except for LSTM-NUE and TRGC (LSTM-NUE, TRGC: Sensitivity $M = 1 \pm 0$ in both dipole conditions), were able to detect this connection in none of the runs or datasets (Table 2).

**Table 2.** Scores of the ANNs in comparison with TRGC, using Ground Truth 2. Results are based upon datasets where all 3 time series (TS) were included as predictors, with one exception: results from Conv2D with two time series as predictors, indicated with *, were also included.

|  | Far-Superficial | | | Far-Deep | | |
|---|---|---|---|---|---|---|
|  | **Sensitivity** | **Precision** | **F1** | **Sensitivity** | **Precision** | **F1** |
| TRGC | **1 ± 0.00** | **0.71 ± 0.29** | **0.79 ± 0.21** | **1 ± 0.00** | **0.70 ± 0.29** | **0.79 ± 0.21** |
| TCDF | 0 ± 0.0 | 0 ± 0.0 | 0 ± 0 | 0 ± 0.0 | 0 ± 0.0 | 0 ± 0 |
| LSTM-NEU | **1 ± 0.0** | **0.37 ± 0.13** | 0.53 ± 0.14 | **1 ± 0** | **0.43 ± 0.09** | 0.60 ± 0.09 |
| CONV2D * (TS=2) | 0 ± 0.0 | 0 ± 0.0 | 0 ± 0.0 | 0.0 ± 0.0 | 0.0 ± 0.0 | 0.0 ± 0.0 |
| Conv2D (TS=3) | 0 ± 0.0 | 0 ± 0.0 | 0 ± 0.0 | 0 ± 0.0 | 0 ± 0.0 | 0 ± 0.0 |

The results of a Scheirer–Ray–Hare Test with method and dipole condition as factors reveal, as expected, a main effect of connectivity method on Sensitivity ($H (4,40) = 47.66$, $p < 0.001$) as well as Precision ($H (4,40) = 48.10$, $p < 0.001$). The interaction between method and dipole condition, nor dipole condition itself were significant (Sensitivity: $H (4,40) = 0.05$, $p = 0.99$, $H (1,40) = 0.01$, $p = 0.91$, interaction and dipole condition effect, respectively; Precision: $H (4,40) = 0.03$, $p = 0.99$, $H (1,40) = 0.00$, $p = 0.95$, interaction and dipole condition effect, respectively). Looking into the effects of different connectivity methods using follow-up Mann–Whitney U tests (Bonferroni-corrected: alpha = 0.05, alpha adjusted = 0.005), significant differences in Sensitivity and Precision between LSTM-NUE/TRGC versus all other methods were found ($p < 0.005$). No differences in Sensitivity between

TRGC (*Mdn* = 1) and LSTM-NUE (*Mdn* = 1) were found (*p* = 0.21), while Precision was significantly higher for TRGC (*Mdn* = 0.67) than for LSTM-NUE (*Mdn* = 0.42), *p* < 0.001). It is not surprising that comparisons of any other ANN method than LSTM-NUE with TRGC were significant, given that these methods had a Sensitivity and Precision of zero. Even though dipole condition did not turn out to exhibit a significant effect on Sensitivity nor Precision in our data sets, this distinction remains theoretically important. We summarized the qualitative differences below.

Sensitivity and Precision were in both dipole conditions 0 while using TCDF and using two different configurations of Conv2D (once with two time series as predictors, once with three time series as predictors). In contrast, Precision was *M* = 0.37 ($\pm$0.13) for LSTM-NUE while TRGC obtained a precision of *M* = 0.71 ($\pm$0.29) in the Far–Superficial dipole condition. In the Far–Deep condition, performance of TRGC remained almost the same (Sensitivity *M* = 1 $\pm$ 0, Precision *M* = 0.70 $\pm$ 0.29) while it became slightly higher (in contrast to the Far–Superficial condition) for LSTM-NUE (Sensitivity *M* = 1 $\pm$ 0, Precision *M* = 0.43 $\pm$ 0.09). F1-scores were *M* = 0.79 $\pm$ 0.21 and *M* = 0.53 $\pm$ 0.14 for TRGC and LSTM-NUE, respectively, in the Far–Superficial condition and *M* = 0.79 $\pm$ 0.21, *M* = 0.60 $\pm$ 0.09 in the Far–Deep condition (while being zero for all the other ANNs).

## 4. Discussion

While considering Sensitivity and Precision, it was shown that, among the ANNs, LSTM-NUE yielded superior results in terms of Sensitivity, resulting in statistically significant differences with the other ANNs except for Conv2D with TS = 2. In terms of Precision, however, no significant differences among the ANNs were found while using Ground Truth 1. TRGC outperformed all ANNs in terms of Sensitivity but, statistically, no differences in Precision were found given that the main effect of the connectivity method was only marginally significant. The lack of a statistically significant effect of connectivity method on Precision, as well as the lack of an effect of dipole condition, and the lack of an interaction effect on both Sensitivity as well as Precision are quite counterintuitive. Indeed, given (1) the patterns observed across both Ground Truths and (2) the results from [1], which convincingly showed effects of different dipole conditions on connectivity patterns as well as interaction effects of connectivity method and dipole condition, one could at least expect an effect of dipole condition. For instance, in [1], it was shown that with an SNR of 0.9 and in a Far–Superficial dipole condition, false positives (as related to Precision) were rather rare, while for other dipole conditions, the percentage of false positives increases (hence decreasing Precision). A related (solely qualitative) observation is the variability in the results of the ANNs (as became obvious through the standard deviations from the mean as depicted in Figures 7 and 8) versus the stability of results produced by TRGC. In particular, ANNS seems to exhibit an increased variability in performance in the Far–Deep Condition (in contrast to the Far–Superficial condition), while almost no such variability is observed for TRGC. A possible culprit could be the initial randomization of the weights in ANNs, but how this instability could differ between architectures or between dipole conditions is unclear and deserves attention in future studies. One of the most important observations of Ground Truth 1 is the relatively poor Precision score of TRGC in the Far–Superficial condition, albeit that a difference with the Far–Deep dipole condition could not be statistically confirmed. More data may be needed to confirm the observed trends. The above-mentioned contrasting results are further discussed below, together with possible explanations with regard to the used connectivity methods.

Using Ground Truth 2, no differences in Sensitivity between TRGC and LSTM-NUE were found given that both methods returned almost always a Sensitivity of one, while Precision was significantly higher for TRGC than for LSTM-NUE. The other ANNs did not detect any connection. The good performance of TRGC regarding Precision is not surprising. In [1], it was already shown that TRGC outperformed Multivariate Granger Causality (MVGC), especially when it comes to false positives (as reflected in a lower False Positive Rate), which is logical given that the introduction of time-reversal could indeed

allow for a better distinction between correlated time series (due to linear mixtures of EEG signals) and true temporal precedence of one time series with regard to another. Although the idea of TRGC is relatively new (as it was first proposed in 2013, by [8]) in comparison to, for instance, bivariate GC and MVGC, due to its appealing theoretical properties as well as its further validation by [7], it was quickly picked up in the field, given its relevance for, among others, EEG source connectivity. Recent developments include, for instance, variations in TRGC that allow for other than normal distributions [32].

In summary, it became clear that, among the ANNs, LSTM-NUE obtained better Sensitivity scores and (although only statistically confirmed using Ground Truth 2) better Precision scores. TRGC outperformed the ANNs in terms of Sensitivity, but in the case of Ground Truth 1, questions arose surrounding its Precision in the Far–Superficial dipole condition (although its Precision was significantly better in Ground Truth 2, without any indication of possible differences between dipole conditions). While all connections were discovered, two false positives were detected relatively consistently, indicating that even with time-reversal there is, in certain circumstances, an over-detection of connections. The lack of performance of TCDF and Conv2D in Ground Truth 2 cannot be due to the location of the two fixed dipoles since they were located at the exact same location as in Ground Truth 1. Hence, we suspect that the moving nature of the sending dipole explains (at least partly) the lack of Sensitivity in TCDF and Conv2D. Taking the results from both Ground Truths together, both LSTM-NUE and TRGC are clearly more sensitive, but they both still tend towards over-detection.

With regard to the score strength rankings, not much can be said about TCDF given that the mean attention scores were significant only for two time series in the Far–Superficial dipole condition, from which one was a falsely detected connectivity (i.e., a false positive). In contrast to TCDF, with LSTM-NUE, for two out of three targets, correct column-wise rankings were obtained for Ground Truth 1. For Conv2D (with TS = 2), correct rankings for predictor pair were found in terms of $R^2$-scores, also for two out of three targets. When looking closer to the contributions of individual time series, it was found that predicting, for instance, X1, with itself and another time series works better than predicting it without the past of X1, which is logical. The fact that adding more predictors (i.e., Conv2D with TS = 3) did not work out is obviously the most problematic aspect of Conv2D. Once a third predictor was added, performance dropped substantially, and it was hypothesized that this could be due to the fact that it was convolving rather uncorrelated or only slightly correlated time series together confuses the two-dimensional network to the extent that no proper prediction can be made. The fact that channels are not kept separate such as in a depthwise-separable architecture, may play an important role in this aspect. Finally, with regard to runtimes (time needed to train a model), LSTM-NUE was together with Conv2D, TS = 3 the most time-consuming method, which calls for a trade-off between accuracy and Time Complexity. It is especially the non-uniform embedding strategy (NUE) that is responsible for the high Time Complexity. However, in [15], it was shown that the current LSTM-model could also produce reasonable results without implementation of the NUE strategy, thereby lowering its Time Complexity drastically.

Moreover, in [15], it was shown that LSTM-NUE could cope with different types of ground truths (linear, non-linear and non-linear with varying length lags), as confirmed in our work. Contrary to [15], we, in addition, had Ground Truth 2 with a moving dipole (i.e., the "Sender") which worked relatively well for LSTM-NUE. Hence, the latter can cope not only with time-varying parameters but also, to some extent, with changing dipole locations. Both TCDF and Conv2D cope far less well with a moving sender, probably (or at least partly) because of the occurrence of both closeness and deepness in the same setting, which has an impact on how signals are transformed by source reconstruction. TCDF and Conv2D are, in contrast to LSTM-NUE, not a part of the family of Recurrent Neural Networks and therefore do not contain feedback loops. The LSTM is particularly known for its excellent memory properties by virtue of its gates that help to remember versus forget certain time samples. In general, the better memory properties of an LSTM

in combination with the NUE approach probably play an important role in dealing with variations over time. An LSTM may also be better in looking through (uncorrelated) noise components because it remembers formerly seen time samples better and, subsequently, should be better in detecting (even weak) patterns over time, also when occluded by noise. This, in turn, may make it easier to deal with more challenging dipole locations or with heavier data transformations. However, this same property could also make an LSTM more sensitive to correlated noise from source mixing. TCDF, on the other hand, has the advantage of a very low Time Complexity, at least partly due to its sparsity in interconnection weights (given its depthwise-separable architecture), but it seems less able to distinguish correlation from causation. This may be due to the lack of feedback loops, an "active" memory feature that makes it difficult to distinguish true patterns from noise over longer time intervals. In this study, TCDF was tuned as such that not too many false positives were detected (given its problem of distinguishing correlation from causation), and this more "conservative" configuration may have led to its low Sensitivity. Overall, we can conclude that, among the ANN models, LSTM-NUE performed best in terms of Sensitivity and Precision regardless of which ground truth was used even though no shuffling or time-reversal was used for connectivity assessment. The contrasting results of TRGC in terms of Precision between dipole conditions in Ground Truths 1 and 2 are puzzling and clearly show an "oversensitivity" of TRGC under certain circumstances. Still, TRGC and LSTM-NUE yielded acceptable-to-good results, albeit both suffer from over-detection. An interesting new finding is the fact that an LSTM is, to some extent, able to provide an answer to the question of whether connectivity between sources is present or absent, at least for source-reconstructed, simulated EEG data. The fact that too many faulty connections were detected (especially in Ground Truth 2) calls for improvements. One possibility is to use LSTM-NUE as part of a masking approach, on top of which another learner is stacked. This masking approach has already led to many advantages in source localization [25], and it may also facilitate connectivity detection with ANNs, especially when overly sensitive to it. In this sense, other ANNs, even with a lower Time Complexity than that of LSTM-NUE, could possibly also be considered as potentially directed connectivity estimators.

An obvious future step is testing whether ANNs can also be applied to real EEG data, albeit that several possible caveats should be taken into account. First and foremost, as shown by [1], under low noise conditions, dipole conditions may matter less, but differences between dipole conditions could become more obvious (i.e., more disturbing) under higher noise levels. Even long-established connectivity methods suffer from this. Since controlling noise levels is hard, reasonably one could opt for EEG-data for which (1) the contributing brain areas are rather superficially located, (2) the connectivity patterns are relatively well known and preferably supported by both high-density EEG and fMRI-data so that a performance evaluation becomes feasible since no ground truth is available for real EEG-data. Testing ANNs and contrasting them with TRGC/other established methods using vision-related or motor-related EEG-datasets makes thus more sense than testing them with data with relatively unknown connectivity patterns. Regions of Interest (ROIs) can be defined based upon previous knowledge about involved brain areas. As for source localization, a reasonable choice is eLORETA. Data-driven approaches (as opposed to ROI-selection), e.g., data-driven clustering [33], seem only reasonable in a later stage when the value of the used ANN is proven on real EEG data.

## 5. Conclusions

Some types of neural networks, in particular LSTMs, may be considered for estimating the directed connectivity of reconstructed EEG Sources. However, no method is flawless, and we showed that even an established method such as TRGC can generate faulty estimates. This calls for further developments. There is much potential for a hybrid approach, in which a neural network could be used as a preprocessing step to chart the

interesting directed connectivity patterns, after which a conventional method is applied for estimating them.

**Author Contributions:** Conceptualization, M.M.V.H., A.F. and I.V.; methodology, I.V. and A.F.; formal analysis, I.V.; resources (scripts), A.F. and I.V.; original draft preparation, I.V.; writing, review and editing, A.F and M.M.V.H.; visualization, I.V.; supervision, A.F. and M.M.V.H.; project administration, A.F. All authors have read and agreed to the published version of the manuscript.

**Funding:** I.V. performed this work as part of her Master in Artificial Intelligence thesis at KU Leuven. A.F. is supported by a grant from the Belgian Fund for Scientific Research—Flanders (FWO 1157019N). M.M.V.H. is supported by research grants received from the European Union's Horizon 2020 research and innovation programme under grant agreement No. 857375, the special research fund of the KU Leuven (C24/18/098), the Belgian Fund for Scientific Research—Flanders (G0A4118N, G0A4321N, G0C1522N), and the Hercules Foundation (AKUL 043).

**Informed Consent Statement:** Not applicable.

**Data Availability Statement:** Not applicable.

**Conflicts of Interest:** The authors declare no conflict of interest. The funders had no role in the design of the study; in the collection, analyses, or interpretation of data; in the writing of the manuscript, or in the decision to publish the results.

## Appendix A

**Table A1.** Scheirer–Ray–Hare Test using Sensitivity as a criterion. Alpha level = 0.05. Significant results shown in bold. Models: Conv2D, TS = 2, kernel size 4 × 2 versus 4 × 4, Conv2D, TS = 3, kernel size 4 × 3 versus 4 × 4, with TS denoting the amount of time series. Model ($H$ (3,32) = 16.04, $p$ = 0.001) and dipole condition ($H$ (1,32) = 4.53, $p$ = 0.03) have significant effects on Sensitivity. Post hoc Mann–Whitney U tests (Bonferroni-corrected alpha level = 0.008) revealed that Sensitivity was significantly higher for Conv2D, TS = 2 (*Mdn* = 0.67, 0.67), versus Conv2D, TS = 3 (*Mdn* = 0.00, 0.17), $p$ = 0.006, $p$ = 0.007 for differing kernel sizes.

| Predictor | Sum of Squares | df | Mean Square | H | *p*-Value |
|---|---|---|---|---|---|
| Dipole condition | 577.60 | 1 | | 4.53 | 0.033 |
| Conv2D model | 2045.35 | 3 | | 16.04 | 0.001 |
| Interaction | 350.15 | 3 | | 2.75 | 0.432 |
| Within | 1999.40 | 32 | | | |
| Total | 4972.50 | 39 | 127.5 | | |

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
