# Peer review of "Neural Networks for Directed Connectivity Estimation in Source-Reconstructed EEG Data"

_applsci, doi:10.3390/app12062889_

Round 1

Reviewer 1 Report

The present manuscript introduces the approach of estimating directed connectivity in source-reconstructed EEG data by means of different types of Artificial Neural Networks (ANNs). Main goal was to introduce these methods, to assess their accuracy with respect to several ground truths and to compare them with already established methods for brain connectivity analyses. In my opinion, on the one hand, an investigation of methods for estimating directed connectivity for source-reconstructed EEG data is of high importance, and on the other hand, the integration of ANNs into this field is of great theoretical and practical relevance. I therefore expressly welcome this approach.

The paper is (up to minor revisions) very well structured and written (in terms of giving the motivation for the study, the theoretical background as well as methodological considerations including the coverage of related literature). Technical/mathematical details of the adapted methods (including the provision of important mathematical formulas), of ground truth data used and the proposed general methodology are well described. In particular, the very consistent specification of the respective configurations for the individual methods should be emphasized here. However, the description and presentation of the results, including their discussion and general conclusions from the study, require at least some revisions.

  1. Most importantly, I think a revision of the discussion and conclusions is needed. So far, the only discussion is that one of the ANN methods (LSTM) can reasonably represent the two ground truths studied (but not better than/equivalent to already established methods such as TRGC). Here, both a more detailed discussion of the problems of the different ANN approaches and a better classification of the TRGC results (including corresponding literature) is needed. Especially the embedding of the reported results in the literature seems to be important (so far there are just two citations of related literature in the whole discussion).
  2. The study investigates (simulated) ground truths data. This is quite useful for the rather detailed investigation of different ANN methods. Nevertheless, an outlook on how to proceed now with the ANN methods for the estimation of directed connectivity (especially with regard to real data) would be important and should be included in the discussion/conclusion section.
  3. Furthermore, throughout the manuscript, a better and more consistent introduction of abbreviations used is needed. For example, abbreviations of methods are introduced in Figure 5 but not in Figure 1 that appears previously. Please check and correct.
  4. The order of the figures in the results section is somewhat confusing. The description starts with Figure 9 and 10 and then jumps to Figure 14. For Figure 8 (which appears first as a graphical representation) I did not find any equivalent in the text at all. Please check and correct this if necessary.
  5. Was normal distribution checked for the application of ANOVA? Please comment on this.
  6. Page 5, line 171: it should be “equation 6” not “equation 7”. Please correct.

Author Response

We would like to thank the reviewer for their careful revision of the manuscript and for their comments that helped us to improve it.

  1. Most importantly, I think a revision of the discussion and conclusions is needed. So far, the only discussion is that one of the ANN methods (LSTM) can reasonably represent the two ground truths studied (but not better than/equivalent to already established methods such as TRGC). Here, both a more detailed discussion of the problems of the different ANN approaches and a better classification of the TRGC results (including corresponding literature) is needed. Especially the embedding of the reported results in the literature seems to be important (so far there are just two citations of related literature in the whole discussion).

The discussion was thoroughly reviewed and restructured. First, a short summary is given of the statistically confirmed results and trends per Ground Truth. This is followed by a discussion of the results with the different ANN methods and TRGC with reference to those  reported in recent literature. Furthermore, since TRGC is a newer variant of Granger Causality, the results were also compared to Multivariate Granger Causality. The discussion ends with future directions for research as explained in our reply to the next comment.

2.The study investigates (simulated) ground truths data. This is quite useful for the rather detailed investigation of different ANN methods. Nevertheless, an outlook on how to proceed now with the ANN methods for the estimation of directed connectivity (especially with regard to real data) would be important and should be included in the discussion/conclusion section.

A future directions section was added at the end of our discussion, in which we provide our view on the application to real EEG-data as well as the inverse method to use, also based on indications from literature.

  1. Furthermore, throughout the manuscript, a better and more consistent introduction of abbreviations used is needed. For example, abbreviations of methods are introduced in Figure 5 but not in Figure 1 that appears previously. Please check and correct.

Abbreviations and symbols were double-checked, and where necessary, corrected throughout the manuscript. Abbreviations for the used methods have been added to the caption of Figure 1.

  1. The order of the figures in the results section is somewhat confusing. The description starts with Figure 9 and 10 and then jumps to Figure 14. For Figure 8 (which appears first as a graphical representation) I did not find any equivalent in the text at all. Please check and correct this if necessary.

Figure- and Table numbering have been revised and references to them checked. The Table regarding several submodels of Conv2D has been moved to the Appendix (A) given that it is less central to the paper than the other reported results.

  1. Was normal distribution checked for the application of ANOVA? Please comment on this.

The ANOVA-assumption of normality had been checked based on kurtosis and skewness, as well as differences between the median and the mean of the corresponding ANOVA-cell groups. Although for Sensitivity most of the groups were approximately normally distributed, this was not so clear for Precision (with mostly Kurtosis exceeding expected values). Therefore, we chose to repeat the relevant analyses using non-parametric tests (Scheirer Ray Hare Test instead of two-way ANOVA and Mann-Whitney U tests, Bonferroni-corrected, instead of Tukey-follow up tests). This was done for: 1) Sensitivity, 2) Precision for both Ground Truths, and 3) the different models from Conv2D as reported in Appendix A. Figures 7 and 8 (on Sensitivity and Precision), and accordingly the result discussion, have been adapted and significances added. Given that these non-parametric tests are rank-based, median values have been added to the corresponding results sections.

This adaptation resulted mostly in changes in the level of Precision (i.e. for Ground Truth 1 no significant differences are found). The results and discussion have been adapted accordingly.

  1. Page 5, line 171: it should be “equation 6” not “equation 7”. Please correct.

Thank you. This mistake was corrected.

Reviewer 2 Report

The Manuscript is devoted to developing ANN-based methods for directed connectivity estimation at the source-level by EEG data. The authors propose 4 ANN-based approaches and test them with 2 model situations with three dipoles each. The work is relevant for neuroscience because reliable methods for connectivity estimation at the source-level are in demand in many areas of research in neuroscience. The work is well written and, at least, of methodological interest.

I have the central question: Do ANN-based methods have any advantages? The results given in the Manuscript show that the conventional method (Time-Reversed Granger Causality) is much more effective.

There are also some inaccuracies and inappropriate phrases in the Manuscript, so careful proofreading is required. For example:

  • Line 15. ”Source-Reconstructed EEG Sources” – repeated word “source”.
  • Line 16. Abbreviations are undesirable in the abstract.
  • Line 60. “.” should be instead “,”.
  • Line 130. The word "Ground" is missing.
  • Line 171. There is no equation (7) in the Manuscript.
  • Figures in the Manuscript should appear in the order of their appearance in the text. There are no references to Fig. 8 in the Manuscript.
  • In line 418, the authors state that ANOVA was used for all ANN models and dipole types. However, in the next paragraph and Fig. 8, only the results for Conv2D are given. The authors should resolve this contradiction.
  • Line 512. The authors state: “Bottom row, black rectangles)”, but there are no black rectangles in Fig. 11.
  • Line 576. “This is a figure” is the mistaken phrase.

Additional recommendations:

  • The coordinates of the dipoles should preferably be given in a standard coordinate system, such as CTF or MNI.
  • Figure 4 should be redrawn in orthogonal cut-views.
  • The authors should explain the reasons for choosing specific configurations of neural networks used.
  • The authors should explain the choice of the cut-off 0.70.
  • The authors should explain the choice of the significance 0.9998.
  • It is recommended to denote significant differences in Figures 9 and 10.
  • Line 465. The authors write: “A two-way ANOVA (with method and dipole condition as factors) revealed”, but there are no results of two-way ANOVA in the Manuscript.

In my opinion, the Manuscript can be accepted for publication after minor changes.

Author Response

We would like to thank the reviewer for their careful revision of the manuscript and for their comments that helped us to improve it.

  1. I have the central question: Do ANN-based methods have any advantages? The results given in the Manuscript show that the conventional method (Time-Reversed Granger Causality) is much more effective.

We started from the question whether ANNs could be useful in the estimation of directed connectivity. Our motivation hereto was two-fold. First, we based ourselves on the literature on Source-Reconstructed EEG activity from which we concluded that directed connectivity estimation after source-reconstruction is still not satisfactory, which is also reflected by the development of new methods or adaptations to older methods. For instance, TRGC, which is in comparison to traditional Granger Causality relatively new, is increasingly becoming popular as it seems to outperform more traditional Granger Causality approaches due to its reliance on time reversal as a validity check. But its performance is still dependent on the context in which it is applied. As a starting point, the simulation study by Anzolin [1] was inspiring because by virtue of their simulation framework several factors were identified that interacted with connectivity estimation. Although a particular method could outperform another, this turned out to be context-dependent.
Second, from a data modeling point of view, ANNs are interesting because they don’t really make assumptions on data distribution nor on signal linearity and, for specific ANNs, even the model order doesn’t have to be pre-specified. This flexibility provides a great advantage to data modelers.

Besides these motivations, at the start of our study, we could not predict whether or not ANNs could challenge TRGC given that a comparison has not yet been done on EEG data.

These motivations have now been stated more clearly in the introduction of the manuscript (page 2), in comparison to the former version of the manuscript.

  1. There are also some inaccuracies and inappropriate phrases in the Manuscript, so careful proofreading is required. For example:

The whole manuscript has been double-checked now, and corrected where necessary.

  • Line 15. ”Source-Reconstructed EEG Sources” – repeated word “source”.

Inaccuracies from lines 15, 60, 130, 171 and 576 have been corrected.

  • Line 16. Abbreviations are undesirable in the abstract.
  • Line 60. “.” should be instead “,”.
  • Line 130. The word "Ground" is missing.
  • Line 171. There is no equation (7) in the Manuscript.
  • Figures in the Manuscript should appear in the order of their appearance in the text. There are no references to Fig. 8 in the Manuscript.

Thank you, the ordering of Figures has been double-checked now and corrected accordingly.

  • In line 418, the authors state that ANOVA was used for all ANN models and dipole types. However, in the next paragraph and Fig. 8, only the results for Conv2D are given. The authors should resolve this contradiction.

The corresponding text was adapted to remove the confusion. For Conv2D, kernel size and number of predictors were checked with ANOVA because Conv2D was not taken from the literature, as we constructed it ourselves, which left us at first with some doubts regarding, among other aspects, the best kernel size. For the other ANNs, parameter tuning was more straightforward which is why we reported the results in the corresponding configuration sections. We have now moved tests regarding kernel size/number of predicting time series regarding Conv2D to Appendix A (Table A1), as it less central in comparison to the Sensitivity and Precision results across methods.

  • Line 512. The authors state: “Bottom row, black rectangles)”, but there are no black rectangles in Fig. 11.

The black rectangles at the bottom of Figure 11 (which is now Figure 9) were deleted because, while taking significance weights into account, there is no significant attention score denoting a directed connection between X2 and X1, nor between X3 and X1. The Figure was adapted (with the rectangles being deleted) but the text not. This is now corrected.

  • Line 576. “This is a figure” is the mistaken phrase. The corresponding caption has been corrected.
  1. Additional recommendations:
  • The coordinates of the dipoles should preferably be given in a standard coordinate system, such as CTF or MNI.

The coordinates of the dipoles have now been transformed into MNI-coordinates, both in Figure 1, as in the manuscript text on page 4.

  • Figure 4 should be redrawn in orthogonal cut-views.

The original Figure was replaced by orthogonal views showing the location of the fixed dipoles using MNI coordinates [mm].

  • The authors should explain the reasons for choosing specific configurations of neural networks used.

The configurations were mostly chosen using a data-driven approach (with the goal of obtaining a good configuration for each ANN in support of a fair comparison), additional explanations have been added at page 6 (top, above Table 1).

  • The authors should explain the choice of the cut-off 0.70. The authors should explain the choice of the significance 0.9998.

Clarification (see also page 9): The used significance level as well as the cutoff for significance weights were experimentally determined and the final choice was based on a data-driven approach by experimenting with significance levels in the range [0.70, 1] and with cutoff-scores in the range [0.40, 0.70].

  • It is recommended to denote significant differences in Figures 9 and 10.

Note that Figure 9 became Figure 7 (on Sensitivity, Ground Truth 1), while Figure 10 became Figure 8 (on Precision, Ground Truth 1). Significant differences are added to Figure 7. Non-parametric analyses using Scheirer Ray Hare Test instead of two-way ANOVA and Mann-Whitney U tests, Bonferroni-corrected, instead of Tukey-follow up tests now replace the analyses done for Sensitivity and Precision. Since no significances were found for Precision, nothing has been adapted in Figure 8.

  • Line 465. The authors write: “A two-way ANOVA (with method and dipole condition as factors) revealed”, but there are no results of two-way ANOVA in the Manuscript.

Instead of ANOVA, to account for non-normality, we now use the Scheirer Ray Hare Test (of which we report the H-statistic and the corresponding p-value) of the effects of method and dipole condition on Precision.

Round 2

Reviewer 1 Report

The authors have made substantial improvements to the manuscript. The discussion has been completely revised and restructured, an outlook for further action is now given, statistical methods used were also revised. The requested minor changes, such as the correct introduction of abbreviations and changes in the order and citation of figures, have also been incorporated. After these changes, I can recommend the manuscript for publication without further revisions.